# Oxidative Stress in Mucopolysaccharidoses: Pharmacological Implications

**DOI:** 10.3390/molecules26185616

**Published:** 2021-09-16

**Authors:** Karolina Pierzynowska, Lidia Gaffke, Zuzanna Cyske, Grzegorz Węgrzyn, Brigitta Buttari, Elisabetta Profumo, Luciano Saso

**Affiliations:** 1Department of Molecular Biology, Faculty of Biology, University of Gdansk, Wita Stwosza 59, 80-308 Gdansk, Poland; karolina.pierzynowska@ug.edu.pl (K.P.); lidia.gaffke@ug.edu.pl (L.G.); zuzanna.cyske@phdstud.ug.edu.pl (Z.C.); 2Department of Cardiovascular, Endocrine-Metabolic Diseases, and Aging, Italian National Institute of Health, 00161 Rome, Italy; brigitta.buttari@iss.it (B.B.); elisabetta.profumo@iss.it (E.P.); 3Department of Physiology and Pharmacology “Vittorio Erspamer”, Sapienza University of Rome, 00185 Rome, Italy; luciano.saso@uniroma1.it

**Keywords:** antioxidants, mucopolysaccharidoses, oxidative stress

## Abstract

Although mucopolysaccharidoses (MPS) are caused by mutations in genes coding for enzymes responsible for degradation of glycosaminoglycans, storage of these compounds is crucial but is not the only pathomechanism of these severe, inherited metabolic diseases. Among various factors and processes influencing the course of MPS, oxidative stress appears to be a major one. Oxidative imbalance, occurring in MPS and resulting in increased levels of reactive oxidative species, causes damage of various biomolecules, leading to worsening of symptoms, especially in the central nervous system (but not restricted to this system). A few therapeutic options are available for some types of MPS, including enzyme replacement therapy and hematopoietic stem cell transplantation, however, none of them are fully effective in reducing all symptoms. A possibility that molecules with antioxidative activities might be useful accompanying drugs, administered together with other therapies, is discussed in light of the potential efficacy of MPS treatment.

## 1. Introduction

The mucopolysaccharidoses (MPS) are a group of rare genetic diseases, being a subset of lysosomal storage diseases (LSD). The disease develops as effects of mutations resulting in the absence or the presence of only low residual activity of lysosomal enzymes responsible for the degradation of glycosaminoglycans (GAG) [1]. Lack or decreased activity of these enzymes leads to GAG storage in lysosomes that eventually leads to progressive damage to cells, tissues, and various organ systems of the body. GAG, also known as mucopolysaccharides, are negatively charged polysaccharide compounds which are components of connective tissue and play an important role in many metabolic processes. Cell signaling processes, including regulation of cell growth, proliferation, promotion of cell adhesion, anticoagulation, and wound repair, represent the main physiological processes that involve GAGs. Mucopolysaccharidoses manifest with variable symptoms depending on the dysfunctional enzyme and associated expression of affected GAG metabolism in organ systems. The GAG accumulation in cells, tissues, and organs throughout the body is considered the main cause of severe symptoms, resulting in loss of independence and premature death of patients [1].

There are 11 types/subtypes of MPS, classified on the basis of the type of stored GAG, and the defect of a specific enzyme (Table 1). Most of them are inherited in an autosomal recessive manner, except MPS II, inheritance of which is X-linked [1].

Types/subtypes of the disease differ in their course. However, all of them are characterized by a number of common symptoms, which include, among others, characteristically coarse facial features, short stature, joint stiffness, hearing and vision disorders, frequent occurrence of hernias, as well as neurological disorders and cognitive impairment that relate to neuronopathic types (Hurler, Hunter, Sanfilippo, and Sly syndromes). Life expectancy is shortened in people with MPS. The main causes of death are systemic or respiratory infections, airway obstruction, and cardiac complications [1].

The most frequently used therapies for MPS include enzyme replacement therapy (ERT), consisting in delivery of the active form of the missing enzyme to the patient’s organism, and hematopoietic stem cell transplantation used in specific cases under multiple conditions [2,3]. However, these therapies are available only for some MPS types (I, II, IVA, VI, and VII). Despite the fact that many additional therapeutic strategies are at the experimental stage (gene therapy, substrate reduction therapy, and others), none of the already known treatments have been able to eliminate all symptoms of the disease [3].

One of the major challenges in developing effective treatment for MPS is the fact that some severe symptoms arise from lesions of hard-to-reach tissues and organs, such as the central nervous system or the skeleton. On the other hand, the existence of additional mechanisms of MPS pathogenesis, in addition to GAG storage, which are poorly understood, has been suggested and demonstrated experimentally [4,5,6,7,8,9,10]. Describing such mechanisms could indicate not only new mechanisms of the pathogenesis of this rare disease, but also the direction to develop effective therapies in the future.

The balance of oxidative processes is extremely important for maintaining the organism’s homeostasis. Its disorders, consisting of excessive accumulation of reactive oxygen species (ROS) or the inability to quickly detoxify reactive intermediates or repair the damage of macromolecules or organelles, have been detected in many human diseases. These include, but are not limited to, neurodegenerative diseases, age-related diseases, depression, autism, respiratory system diseases, cardiovascular diseases, and cancer [11,12,13,14,15]. Although mentioned in various reports and review articles on MPS, oxidative stress has not been, however, considered to a large extent as one of the crucial mechanisms contributing to or modulating the pathogenesis of MPS. Moreover, not many compounds with antioxidant activity have so far been tested as potential drugs for MPS.

In this critical review, we collect the previously described data about oxidative imbalance in MPS and discuss the results of research on the activity of antioxidants in the search for drugs for this disease, used separately or as adjuvants.

## 2. Oxidative Stress as One of the Mechanisms of MPS Pathogenesis

Oxidative stress has been detected in many human diseases. Disturbances in oxidative processes may have the following effects: (i) they contribute to neuronal loss (in cerebral ischemia, seizure disorders, schizophrenia, Parkinson’s disease, and Alzheimer’s disease) [12], (ii) they promote prothrombotic reactions (in diabetes), (iii) they increase the level of oxidized cholesterol (in cardiovascular diseases), (iv) they cause glomerular damage and renal ischemia (in acute and chronic kidney disease), (v) they induce immunological responses and chronic inflammation (in chronic obstructive pulmonary disease, depression, and cancer) [13,15], and (vi) they result in mitochondrial dysfunctions (together with impaired immune response and iron and copper metabolism in autism) [11]. In most of these diseases, it is not yet clear whether the oxidative stress phenomenon is the result of disturbed cell homeostasis or one of its causes. Many of these features are also characteristic for MPS, and they were commonly assessed as important contributions to neurodegeneration (in MPS I, II, III, or VII), and also less frequently described as mitochondrial dysfunction [16,17]. However, there are only relatively few reports on oxidative stress in lysosomal storage diseases.

The first observations on oxidative imbalance were reported by Simonaro’s group in MPS type VI, a disease characterized by an abnormal cartilage and bone development, short stature, dysostosis multiplex, cranial and tracheal abnormalities, cornea and cardiovascular defects, and degenerative joint disease [18]. In an animal model of MPS VI, the authors observed an age-dependent increase in chondrocyte apoptosis within synovial joint defects and defects in the articular cartilage matrix composition, including a generalized proteoglycan and collagen depletion. In ex vivo experiments, cultured MPS VI articular chondrocytes released increased levels of nitric oxide (NO) and tumor necrosis factor alpha (TNFα) compared to control healthy cells. In addition, the in vitro treatment with dermatan sulphate induced the release of increased amounts of NO in pathological and healthy cells, thus proving the important role of this GAG in the abnormal cartilage matrix homeostasis [18]. The research on NO has been continued, in which the role of NO in the phenomenon of increased expression of matrix-degrading metalloproteinases (MMPs) was proposed, which resulted in increased chondrocyte proliferation as well as a reduction in chondro-differentiation in the growth plates demonstrated in the endochondral ossification studies on the rat model MPS VI [19]. Therefore, these studies show a faster growth of MPS chondrocytes, but a lack of maturity and mineralization in the skeleton. The authors indicated NO as one of the important factors in the pathogenesis of MPS, which could be a diagnostic marker or a therapeutic target for this disease.

The possible involvement of ROS in the genesis of neurodegeneration in MPS was proved later, by analyzing the RNA and/or protein levels for some components of the phagocytic enzyme complex NADPH-dependent oxidase (p47phox, p67phox, gp91phox) as well as other genes involved in oxidative reactions (Sod1, Ret, Bmp4, Tgfb, Gzmb i Prf1) in the MPS IIIB mice brains [20,21]. In one month-old-mice, at least three components of this complex were upregulated and, more importantly, this upregulation resulted in an increased NADPH oxidase activity. Time-course data indicate that oxidative stress is already present in young mice (one month old) and that it results primarily in protein oxidation (high levels of protein carbonyl (CO) groups), while lipid peroxidation (malondialdehyde (MDA) and 4-hydroxynonenal (4-HNE)) and DNA oxidation (8-hydroxy-2′-deoxyguanosine (8-OHdG)) appear milder and essentially restricted to the cerebellum [21]. Similar parameters related to oxidative stress were also assessed in the mouse MPS I model (*Idua*^−/−^). Indeed, both in the tissues of the nervous system and in the peripheral organs, significant increases in the activity of superoxide dismutase (SOD) and catalase as well as an increased number of carbonyl groups were measured [22].

As mentioned previously, it is not entirely clear whether oxidative stress is an independent cause or a consequence of disturbed cell homeostasis in the context of human disease. A possible link between early neuroinflammation and oxidative stress in patients with MPS is represented by GAGs degradation products, which being structurally similar to lipopolysaccharide (LPS)—an endotoxin of Gram-negative bacteria product—may bind and activate the Toll-Like Receptor 4 (TLR4), thus leading to pro-inflammatory cytokine release and innate immune system activation [23]. Trudel et al. used the double mutant TLR4 knockout MPS IIIB mice (*Naglu*^−/−^ mice with an additional knockout in the tlr4 gene), which showed no neuroinflammation, whereas inflammation was present in the cortex of 10-day-old MPS IIIB mice [24]. By analyzing the total SOD and glutathione peroxidase (GPx) activities in the brain of MPS IIIB mice and double mutants, a small antioxidant response was observed at an early stage, even in the absence of inflammation. These findings suggest that oxidative stress is not a consequence of the inflammation of the nerve cells in MPS IIIB, but it might be one of the primary causes of cellular dysfunction during neurodegeneration.

The results obtained in animal models were also confirmed in clinical trials. Clinical investigations in MPS IVA patients showed a significant increase in GPx activity in erythrocytes along with an increase in reduced glutathione (GSH) levels, while SOD and glutathione reductase (GR) activity remained similar to the control group. In urine samples of patients with MPS IVA, increased levels of 15-F2t-isoprostane and 8-OHdG were also identified, indicating increased intensity of lipid peroxidation and DNA oxidative damage without any changes in proteins’ oxidation (as measured by the level of sulfhydryl groups) [25].

Research on the role of oxidative stress in the pathogenesis of MPS also included two commonly used antioxidants, melatonin and coenzyme Q10 (CoQ10). Melatonin is known as a hormone that regulates the human circadian rhythm. However, this hormone is also the antioxidant of the body’s first line of defense against damage caused by oxidative stress. Studies carried out on patients with MPS III showed disturbances in the daily production of melatonin, which was associated in the first place with sleep disorders, especially characteristic of Sanfilippo syndrome [26,27]. However, it is also worth considering the possible role of melatonin in changes in oxidative balance, which has not been proposed previously. Studies in MPS patients have also shown decreased plasma levels of CoQ10 along with decreased levels of pyridoxal phosphate (PLP) in plasma and cerebrospinal fluid (in some patients) [28,29].

It is worth noting that protein damage caused by ROS was detected in MPS [21,22], especially in patients on enzyme replacement therapy [30], however, other studies failed to detect protein damage and excluded an involvement of the unfolded protein response (UPR) of the endoplasmic reticulum (ER) in the genesis of MPS [25,31]. Therefore, one might suggest that this discrepancy could be ascribed to the model used or to the clinical setting related to the therapy. It also should be noted that some proteins belonging to the PDI family have long been known to have, besides the protein disulfide isomerase and chaperone activity, cystein protease activity, and are potential candidates for the ER proteolytic machinery. Indeed, the expression of Pdia5 resulted upregulated in MPS fibroblasts, thus suggesting that its upregulation could be an early event of ER stress possibly related to the severity of the damage induced in the mutant proteins [31]. Further studies are needed to clarify whether UPR mechanisms are not functioning properly in MPS cells, thus contributing to inflammation and oxidative stress in MPS diseases.

A summary of the results of studies on oxidative stress in MPS, published to date, is presented in Table 2.

## 3. Enzyme Replacement Therapy and Oxidative Stress in MPS Patients

One question that should be asked is whether the oxidative stress observed in MPS is directly related to GAG storage. If yes, the observed pathogenesis mechanism should potentially be eliminated after applying the most common therapy for MPS, which is ERT (the use of which leads to a reduction in GAG levels). However, if not, it would suggest the need to include antioxidants as additional compounds that can aid ERT in disease types where this therapy is not fully effective.

The first studies on the influence of ERT on the markers of oxidative stress concerned the level of antioxidant enzymes such as SOD and catalase. Clinical trials conducted in patients with MPS I showed an increase in catalase activity after 4 weeks of therapy and a decrease in SOD activity after 12 weeks of therapy. The level of GSH was also analyzed as one of the indirect indicators of GPx activity. However, it was unchanged after using ERT (Aldurazyme^®^ (laronidase), at the dose of 0.58–1.8 mg/kg) for 24 weeks [36]. Unfortunately, it is not known whether the detected changes in SOD and catalase activity are initially increased or decreased compared to healthy people due to the lack of such a group in the experimental design of this study.

Most studies on the influence of ERT on the oxidative profile have been carried out in patients with MPS II. The first of these studies showed no changes in catalase and SOD activity under the influence of ERT during the first 6 months of treatment (ERT treatment followed the protocol of the Medical Genetic Service of the Clinical Hospital of Porto Alegre, RS, Brazil), despite an elevated baseline catalase activity relative to the healthy control group. On the other hand, the level of MDA (a marker of lipid peroxidation) decreased, and the level of sulfhydryl groups increased (both parameters initially increased in patients with MPS II compared to the control group) [32]. A decrease in elevated levels of MDA was also reported in two other studies involving MPS II patients treated with ERT using both Elaprase^®^ (idursulfase) at 0.5 mg/kg weekly for 6 months [34] and following the protocol of the Medical Genetic Service of the Clinical Hospital of Porto Alegre, RS, Brazil, also for 6 months [33]. The latter study also reported elevated levels of carbonyl groups in the plasma of patients before therapy, indicating intensified processes of protein oxidation, as well as significantly increased DNA damage (determined by the comet assay) compared to healthy people. These effects after ERT application were alleviated, but did not reach the control values (recorded in healthy people). More recently, a clinical study conducted in patients affected by Fabry disease (FD), characterized by the deficiency of the enzyme α-galactosidase A and the accumulation of globotriaosylceramide, showed that ERT was not able to reverse the higher lipid peroxidation and urinary NO**^•^** equivalents levels observed in patients when compared to healthy subjects, thus demonstrating that redox imbalance still occurs after ERT and suggesting the possible usefulness of antioxidants in combination with ERT as a more effective therapeutic approach [30].

The effects of long-term ERT for MPS II patients (mean duration of 5.2 years) (Elaprase^®^ at 0.5 mg/kg) have also been studied. In these patients, no changes in the activity of antioxidant enzymes, glutathione levels, or sulfhydryl and carbonyl groups in plasma were observed after the enzyme administration [35]. It is possible that this is a normalization effect under the influence of therapy, but it is difficult to prove this due to the lack of data before the start of the therapy. These patients, however, were characterized by elevated levels of 15-F2t-isoprostane in the urine and substances reactive with thiobarbituric acid in the plasma (markers of lipid peroxidation), as well as increased concentrations of di-tyrosine in the urine (markers of protein oxidation) [35]. Subsequent follow-up of this study (ERT treatment lasting for 5.7 years on average) showed increased levels of 8-OHdG, 8-hydroxyguanosine, and 8-hydroxyguanine in urine, indicative of oxidative damage to nucleic acids despite ongoing long-term therapy [37].

Changes in the oxidative status of the body were also investigated in MPS IVA patients undergoing 32 weeks of ERT, on average (Vimizim^®^ (elosulfase alfa) at 2 mg/kg weekly) [38]. These studies indicated a reduced content of GSH and an increased activity of SOD in erythrocytes. There were also increased levels of isoprostanes and dityrosine in the urine, and decreased levels of sulfhydryl groups, as well as increased levels of 8-OHdG in the plasma compared to the control group [38]. These results indicated damage to cell components caused by oxidation reactions despite taking the enzyme.

## 4. Effects of Experimental GAG Level-Decreasing Therapies on Oxidative Stress in MPS

In addition to ERT research, experimental studies with other therapeutic strategies also aimed at lowering GAG levels in MPS. One of them was gene therapy, tested in preclinical and clinical studies. The ability of different viral vectors to ameliorate phenotypical alterations have been evaluated in MPS animal models [39]. Studies performed using the MPS IIIB mouse model have shown that a lentiviral vector carrying the functional human *NAGLU* gene, delivered to the brain of mice, led not only to an increase in the activity of the enzyme resulting in the improvement of behavioral disorders and a decrease in GAG levels, but also to a decrease in the expression of genes related to inflammation and oxidative stress (gp91phox) [40]. Furthermore, administration of the gene vector AAVrh10-SGSH-IRES-SUMF1 to correct sulfamidase deficiency in a mouse model of MPS IIIA has been demonstrated to be able to improve GM3 ganglioside accumulation, and to reduce microglial activation and ubiquitin-positive lesion formation [41]. The intracerebral administration of viral vectors as a therapeutic approach for MPS has also been used in humans. In particular, a phase I/II clinical trial has been performed in four children affected by MPS IIIA who were treated with the viral vector rh.10-SGSH-IRES-SUMF1 administered in the white matter by intracerebral injections [42]. During the year of follow-up, the authors observed that the treatment was well-tolerated by the patients and no adverse events were recorded. Neuropsychological evaluations indicated a moderate amelioration of behavior, attention, and sleep in 3 out of 4 patients [42].

As the intracerebral administration is highly invasive, other delivery systems such as intravenous injection have been evaluated. Viral vectors with the ability to target the liver and carrying liver-specific promoters have been developed to make the liver capable of producing large amounts of the deficient lysosomal enzymes. This approach has been used in MPS I and MPS VI feline animal models [43,44]. Nevertheless, the presence of the blood–brain barrier (BBB) represents a difficult obstacle to overcome for the use of this approach.

In conclusion, although encouraging results have been obtained in pre-clinical and clinical studies, further studies are needed to develop effective therapies for MPS patients’ management. Therefore, the use of antioxidants in assistance to the major therapy for MPS seems to be not only a beneficial, but perhaps also a necessary strategy.

## 5. Compounds with Antioxidant Activities and Their Effects in MPS

Most of the studies carried out before 2000 on antioxidants in MPS were related to the use of vitamins, and the purpose of these studies was mainly to investigate their effects on GAG levels. Experiments with the use of ascorbic acid (vitamin C) (at 50 µg/mL) showed an increase in the level of heparan sulfate in cultured fibroblasts from MPS I patients. This phenotype was not observed in the case of healthy fibroblasts [45,46]. Different results were obtained with the use of retinol (vitamin A alcohol). In contrast to ascorbic acid, retinol (50 IU/mL of medium) reduced the total GAG content in cultures of fibroblast of various types of MPS [47]. Similar results were obtained after 10 years of vitamin A supplementation in a patient with Hurler-Scheie syndrome (daily doses of 1000 to 2000 IU/kg). However, in addition to decreased urinary GAG levels, side effects such as irritability, bone pain, dizziness, vomiting, and diarrhea were also observed [48]. However, these results are in contrast with those obtained in a study performed in a larger group of patients with MPS I and II taking vitamin A for 4–14 months (4000 IU/kg). That study showed no change or an increase in urinary GAG levels, as well as hepatosplenomegaly and neurological regression [49].

A more recent study on MPS with the use of vitamins concerned vitamin E. Zalfa et al. [50] observed that in the MPS II (*Ids*^−/−^) mouse model, neurodegeneration is preceded by extensive inflammation. In search of the mechanism of this phenomenon, they indicated that neural stem cells (NSCs) derived from the *Ids*^−/−^ mice died significantly more often when cocultured with healthy neurons. This effect was not observed under conditions of low oxygen or vitamin E concentrations, and therefore, the authors proposed that the oxidative stress led to neuroinflammation, which subsequently caused nerve cell death in MPS II [50]. Still, attention is paid to conflicting research results regarding the use of vitamins in anti-MPS therapies, obtained by different research groups [51].

In vitro studies on normal (non-MPS) Chinese hamster ovary cells showed a large decrease in the synthesis of GAG (cell-bound and released to agar) under the influence of one of the organic compounds from the group of sulfoxides, dimethyl sulfoxide (DMSO) [52]. The experiments carried out many years later on fibroblasts from healthy donors and from patients with MPS IIIA and IIIB confirmed a moderate impact of DMSO on GAG synthesis [53]. Modulation of the expression of genes involved in GAG synthesis has been proposed as a mechanism for this phenomenon. It was observed that the positively regulated transcripts are involved in glycan biosynthesis and metabolism, while the negatively regulated transcripts are involved in lysosomal metabolism and autophagy inhibition [53].

Many non-vitamin antioxidant compounds have been tested in MPS using their various biological activities (not necessarily oxidative stress). This was the case with melatonin and genistein. Melatonin was administered to patients with Sanfilippo syndrome to effectively regulate sleep in patients. Taking into account the disturbance of its daily production in these patients, it is not surprising that melatonin’s beneficial effects were observed [54]. Although the antioxidant properties of melatonin lead to neuroprotection [55], alleviation of hypertension [56], or regulation of the activity of natural killer cells [57], this hormone has already been used only in some human diseases, and it has never been studied in MPS in light of oxidative stress prevention.

Genistein, one of the isoflavones, is also known for its antioxidant properties. It has been shown to reduce the levels of ROS and MDA, to increase the levels of antioxidant enzymes, and to be beneficial as a protective agent against the damage to mitochondria (as organelles that generate the highest amounts of ROS). These features have been used in experimental therapies for Alzheimer’s disease [58,59,60], Parkinson’s disease [61], and Huntington’s disease [62]. However, in the case of MPS I, II, and III, the focus was on a different mechanism of its action, namely inhibition of the autophosphorylation of the epidermal growth factor receptor (EGFR). This resulted in the inhibition of the expression of genes encoding GAG synthetases, leading to a decrease in their accumulation in fibroblasts collected from patients [63,64]. Moreover, genistein-mediated stimulation of lysosomal biogenesis through the TFEB-dependent pathway was demonstrated [65], suggesting that elevated activities of lysosomes, and perhaps enhanced autophagy, may contribute to more effective removal of GAG deposits. These results were confirmed in the MPS IIIB mouse model (genistein was administered at the dose of 160 mg/kg/day), indicating a decrease in GAG levels in peripheral organs and the central nervous system, reducing inflammation and contributing to the elimination of behavioral disturbances [66,67]. Subsequently, additional mechanisms of genistein action were proposed in the context of ameliorating the pathogenesis of MPS. Regulation of the cell cycle [68] and regulation of proteasome activity [69] were reported. These studies, however, did not determine the influence of genistein on the status of oxidative stress, and it is not known to what extent this property contributes to the improvement of the disease phenotype.

Moreover, Fumić et al. [70] proposed a novel MPS therapy using compounds of plant origin. They developed an efficient chemical process to obtain a Medicago sativa extract using ultrasound extraction and hydroxypropyl-β-cyclodextrin complexation. The extract obtained by these authors showed antioxidant and iron ions’ chelating properties, and was able to reduce GAG levels in skin fibroblasts obtained from a patient with MPS III. At very low concentrations, 3 and 6 μg/mL, the reduction observed was of 41.2% and 51.1%, respectively [70]. However, as with melatonin and genistein, the effects on the oxidative status of cells have not been studied.

Recent reports on the use of phytotherapeutic agents focused on one of the grape compounds, resveratrol. Studies carried out on the *Drosophila melanogaster* model of MPS VII (created by knocking out the CG2135 gene, coding for the fly β-glucuronidase orthologue) indicated attenuation of neuromuscular degeneration (determined on the basis of increased number of dopaminergic neurons) and restoration of normal motor function (estimated on the basis on fewer apoptotic cells in the muscles and restoration of the ability of flies to climb) [71]. The use of many properties of resveratrol, including antioxidant activities, has already been proposed in the treatment of neurodegenerative diseases, liver diseases, cardiovascular diseases, and rheumatology diseases. Recent studies, however, draw attention to the possibility of autophagy induction by resveratrol which, together with its antioxidant properties, could be successfully used in the treatment of MPS [72].

Interestingly, some research groups specifically targeted the antioxidant properties of the tested compounds in MPS treatment. This applies to aspirin, CoQ10, and selenium. Long-term use of aspirin has been tested in the treatment of MPS III [73]. The authors pointed to genes related to oxidative stress (iNOS, p47phox, p67phox, sod2, gpx4), whose expression was upregulated, correlating with extensive neurodegeneration in the nervous tissues of MPS IIIA mice. Application of aspirin at 200 mg/kg, through intraperitoneal injection, three times a week, normalized the mRNA levels of inflammation- and oxidative stress-related genes in mouse brains, and had a positive effect on GPx activity (measured as the ratio of reduced GSH to its oxidized derivative) [73].

Matalonga et al. [74] obtained interesting results when testing the effects of CoQ10 and a cocktail of antioxidants (α-tocopherol, N-acetylcysteine, and α-lipoic acid) on enzymatic activity and GAG levels in fibroblasts from five MPS III patients. Their results indicated that CoQ10 supplementation (50 μmol/L) increased the residual enzyme activity in two cell lines taken from patients with MPS IIIB. CoQ10 supplementation, as well as a cocktail of antioxidants, also increased the efficiency of exocytosis and decreased the level of GAG accumulation in three cell lines obtained from patients with MPS IIIA and IIIB [74].

Selenium was also tested in MPS for its antioxidant properties. Its influence on antioxidant processes was studied in patients with MPS I, II, and VI undergoing ERT. In these patients, decreased levels of this element were found, indicating a high selenium deficiency. Selenium supplementation for six months was associated with a decrease in GSH and GSH/GSSG ratio, as well as a decrease in GPx activity as compared to the time before supplementation, suggesting that these patients could benefit from selenium supplementation [75].

## 6. Concluding Remarks

The role of oxidative stress in the development of the pathogenesis of MPS has been indicated. Both participation of GAG in oxidative stress induction and the possible use of antioxidants as adjuvants in the treatment of MPS can be postulated on the basis of reported studies. However, specific mechanisms of influence of oxidative stress on MPS pathology, as well as efficiency of antioxidants in reduction of MPS symptoms, remain to be further investigated.

## Figures and Tables

**Table 1 molecules-26-05616-t001:** Characteristics of types/subtypes of MPS.

MPS Type	MPS Eponyme	Defective Enzyme	Stored GAG(s)	OMIM No.
MPS I	Hurler syndrome, Hurler–Scheie syndrome, Scheie syndrome	α-L-iduronidase	heparan sulfate dermatan sulfate	252800/607014, 607015, 607016
MPS II	Hunter syndrome	2-iduronate sulfatase	heparan sulfate dermatan sulfate	300823/309900
MPS IIIA	Sanfilippo syndrome A	N-sulfoglucosamine sulfhydrolase	heparan sulfate	605270/252900
MPS IIIB	Sanfilippo syndrome B	α-N-acetylglucosaminidase	heparan sulfate	609701/252920
MPS IIIC	Sanfilippo syndrome C	Acetyl-CoA:α-glucosaminide acetyltransferase	heparan sulfate	610453/252930
MPS IIID	Sanfilippo syndrome D	N-acetylglucosamine 6-sulfatase	heparan sulfate	607664/252940
MPS IVA	Morquio syndrome A	N-acetylgalactosamine 6-sulfatase	keratan sulfate chondroitin sulfate	612222/253000
MPS IVB	Morquio syndrome B	β-galactosidase-1	keratan sulfate	611458/253010
MPS VI	Maroteaux–Lamy syndrome	N-acetylgalactosamine 4-sulfatase	dermatan sulfate	611542/253200
MPS VII	Sly syndrome	β-glucuronidase	heparan sulfate, dermatan sulfate, chondroitin sulfate	611499/253220
MPS IX	Natowicz syndrome	Hyaluronidase-1	hyaluronic acid	607071/601492

**Table 2 molecules-26-05616-t002:** Markers of oxidative stress in MPS.

Oxidative Stress Marker	MPS Type	Used Model	Results	Reference
Modulation of levels of simple inorganic compounds	VI	Rat and cat articular chondrocytes	↑ NO	[18]
		ASB activity-deficient rats	↑ NO	[19]
Lipid peroxidation	II	Patients (plasma)	↑ MDA	[32,33,34]
	IIIB	*Naglu*^−/−^ mice	↑ MDA and 4-HNE	[21]
	IVA	Patients (urine)	↑ 15-F2t-isoprostane	[25]
Protein oxidation	I	*Idua*^−/−^ mice	↑ Carbonyl groups	[22]
	II	Patients (plasma)	↑ Sulfhydryl groups	[32]
			↑ Carbonyl groups	[33]
	IIIB	*Naglu*^−/−^ mice	↑ Carbonyl groups	[21]
DNA oxidation(and damage)	II	Patients (plasma)	↑ DNA damage (assessed by comet assay)	[33]
	IIIB	*Naglu*^−/−^ mice	↑ 8-OHdG	[21]
	IVA	Patients (urine)	↑ 8-OHdG	[25]
Modulationof activities ofantioxidantenzymes	I	*Idua*^−/−^ mice	↑ SOD and catalase	[22]
	II	Patients (erythrocytes)	↑ catalase	[32]
	IIIB	*Tlr4*^−/−^*Naglu*^−/−^ mice	↑ SOD and GPx	[24]
	III	Patients (urine)	↑ Melatonin (day) ↓ Melatonin (night)	[26,27]
		Patients (plasma)	↓ Coenzyme Q10	[28]
	IVA	Patients (erythrocytes)	↑ GPx	[25]
			↑ SOD	[35]
Gene expression modulation	IIIB	*Naglu*^−/−^ mice	↑ *p47^phox^*, *p67^phox^*, *gp91^phox^*, *Sod1*, *Ret*, *Bmp4*, *Tgfb, Gzmb*, *Prf1*	[21]

Abbreviations: 4-HNE, 4-hydroxynonenal; 8-OHdG, 8-Hydroxy-2’-deoxyguanosine; ASB, arylsulfatase B; GPx, glutathione peroxidase; MDA, malondialdehyde; NO, nitric oxide; SOD, superoxide dismutase; ↑, an increase; ↓, a decrease.

## Data Availability

Not applicable.

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
