# Peer review of "Oxidative Stress in Mucopolysaccharidoses: Pharmacological Implications"

_molecules, 2021, doi:10.3390/molecules26185616_

Round 1
Reviewer 1 Report
This paper is a splendid review of oxidative stress in mucopolysaccharide disorders and highlights the need for adjunct or alternative treatments for these lysosomal storage disorders. The only comment I have is that there is no mention of similar studies performed in canine models of MPS (Sun Peck et al). Of minor concern: there are a few typos that need to be addressed.
Reviewer 2 Report
Dear Authors,
congratulations for the topic you describe that is very fascinating and intriguing as well as current. You describe well the phenotipic continuum that characterize the MPS group. The reading appear sometimes disconnetted and many concepts are not well linked each other in the discussion. I appreciapte the discussion about neuroinflammation: the review and analysis are well conducted. Many points need deepening and multicenter study. The table 2 is very usefull to summarize the text.
At the point 3 the text appear too much technical in some passages and the part of therapeutical approach need to be linked with the rest of the text: it would be better to reserve a different point for this point. Do you conduct studies about burden disease in patients who underwent to HSCT?
Author Response
As suggested by the reviewer, section 3 has been divided into two sections, separating already used enzyme replacement therapy and experimental therapies, and their effects on oxidative stress. Different parts of the text are now linked. No data related to effects of HSCT on oxidative stress in MPS are available, and we did not conduct such studies.
Reviewer 3 Report
The article entitled " Oxidative stress in mucopolysaccharidoses: pharmacological implications" is a well-presented and comprehensive summary of the most important information and scientific achievements concerning the role of oxidative stress in pathogenesis of mucopolysaccharidoses and the influence of antioxidant therapy on the markers of oxidative stress. In my opinion, a review is suitable for publication in Molecules. Below minor technical comments, which should be corrected prior to publication:
- line 99 – double space between “MPS” and “type”
- line 104 – “ex vivo” should be in italics
- line 123 – double space between “activity.” and “Time-course”
- line 126 – should be (4-HNE) instead of (4- HNE)
- line 209 – double space between “MDA” and “was”
- line 345 – “Drosophila melanogaster” should be in italics
Author Response
All technical comments were addressed, and all corrections have been introduced into the text.